# Compartmentalization and persistence of dominant (regulatory) T cell clones indicates antigen skewing in juvenile idiopathic arthritis

Gerdien Mijnheer[1†], Nila Hendrika Servaas[1†], Jing Yao Leong[2], Arjan Boltjes[1], Eric Spierings[1], Phyllis Chen[2], Liyun Lai[2], Alessandra Petrelli[1], Sebastiaan Vastert[1,3], Rob J de Boer[4], Salvatore Albani[2], Aridaman Pandit[1*‡], Femke van Wijk[1*‡]

[1]Center for Translational Immunology, University Medical Center Utrecht, Utrecht University, Utrecht, Netherlands; [2]Translational Immunology Institute, Singhealth/Duke-NUS Academic Medical Centre, the Academia, Singapore, Singapore; [3]Pediatric Immunology & Rheumatology, Wilhelmina Children's Hospital, University Medical Center Utrecht, Utrecht University, Utrecht, Netherlands; [4]Theoretical Biology, Utrecht University, Utrecht, Netherlands

*For correspondence:
A.Pandit@umcutrecht.nl (AP);
F.vanWijk@umcutrecht.nl (FW)

[†]These authors contributed equally to this work
[‡]These authors also contributed equally to this work

Competing interest: The authors declare that no competing interests exist.

**Abstract** Autoimmune inflammation is characterized by tissue infiltration and expansion of antigen-specific T cells. Although this inflammation is often limited to specific target tissues, it remains yet to be explored whether distinct affected sites are infiltrated with the same, persistent T cell clones. Here, we performed CyTOF analysis and T cell receptor (TCR) sequencing to study immune cell composition and (hyper-)expansion of circulating and joint-derived Tregs and non-Tregs in juvenile idiopathic arthritis (JIA). We studied different joints affected at the same time, as well as over the course of relapsing-remitting disease. We found that the composition and functional characteristics of immune infiltrates are strikingly similar between joints within one patient, and observed a strong overlap between dominant T cell clones, especially Treg, of which some could also be detected in circulation and persisted over the course of relapsing-remitting disease. Moreover, these T cell clones were characterized by a high degree of sequence similarity, indicating the presence of TCR clusters responding to the same antigens. These data suggest that in localized autoimmune disease, there is autoantigen-driven expansion of both Teffector and Treg clones that are highly persistent and are (re)circulating. These dominant clones might represent interesting therapeutic targets.

## Editor's evaluation

In this study, the authors performed mass cytometry analysis and T cell receptor sequencing to study different joints affected at the same time and over the course of the relapsing-remitting disease and found that the composition and functional characteristics of immune infiltrates are similar between joints within one Juvenile Idiopathic Arthritis patient. They observed an overlap between dominant T cell clones, especially Treg, and some of these clones could be detected in circulation and persisted over the course of the relapsing-remitting disease. They demonstrated that these T cell clones were characterized by a high degree of sequence similarity, indicating the presence of TCR clusters responding to the same antigens.

## Introduction

Inflammation, often localized to specific target tissues, is a hallmark of autoimmune diseases. In these diseases, multiple sites within specific tissues can be inflamed in tandem. An example of this phenomenon includes the inflammation of multiple joints in juvenile idiopathic arthritis (JIA). Multiple lines of evidence implicate T cells as key players of this tissue-specific autoimmune inflammation. First, many autoimmune diseases are associated with the expression of specific MHC (HLA) class II alleles, which is hypothesized to lead to altered antigen presentation and enhanced CD4+ T cell activation (*David et al., 2018*). Second, activated CD4+ T cells often accumulate in affected tissue (*Black et al., 2002*). Finally, CD4+CD25+CD127lowFOXP3+ regulatory T cells (Tregs), capable of suppressing immune responses and fundamental to immune homeostasis, also accumulate in the affected tissue (*Wehrens et al., 2013*; *Long and Buckner, 2011*).

Tissue-resident T cells display an array of distinct trafficking and functional markers compared to circulating T cells (*Kumar et al., 2017*; *Nistala et al., 2010*; *Cosmi et al., 2011*; *Ohl et al., 2018*; *Wehrens et al., 2011*; *Duurland et al., 2017*). Novel technologies, such as mass cytometry (CyTOF), allow for high-resolution analysis of the cellular heterogeneity within inflamed tissues to reveal potential pathogenic T cell populations. Moreover, studies assessing the T cell receptor (TCR) repertoire have generated evidence for the presence of clonally expanded T cells in specific tissues in autoimmune diseases (*Muraro et al., 2006*; *Chapman et al., 2016*; *Doorenspleet et al., 2017*; *Günaltay et al., 2017*; *Musters et al., 2018*). These findings suggest that tissue-specific T cell responses are mounted by specific local antigens that selectively induce activation, expansion and/or migration of antigen-specific T cell clones.

Similar to conventional T cells, Tregs that leave the thymus typically express a unique TCRs. While Tregs only represent a small fraction of the total CD4+ T cell pool, the TCR repertoire of peripheral Tregs is as diverse as that of conventional CD4+ T cells (*Wing and Sakaguchi, 2011*; *Leung et al., 2009*; *Wang et al., 2010*). Several studies previously showed that a restricted TCR repertoire of the Treg compartment can lead to the development of autoimmune disease (*Adeegbe et al., 2010*; *Föhse et al., 2011*; *Yu et al., 2017*; *Nishio et al., 2015*). However, Tregs with a single TCR specificity can also inhibit autoimmune responses, thereby also providing some degree of protection against autoimmunity (*Levine et al., 2017*). In JIA, hyper-expanded Treg TCRβ clones can be found at the site of inflammation (*Bending et al., 2015*; *Rossetti et al., 2017*; *Henderson et al., 2016*), and in refractory JIA patients hyper-expanded Tregs can even be found in circulation (*Delemarre et al., 2016*). This expansion is likely caused by a dominance of specific (auto)antigens present at target tissues. However, the exact antigen specificity and temporal and spatial dynamics of hyper-expanded effector T cells and Tregs in chronic inflammation and their relation to disease relapses remain to be established. Defining the specific CD4+ T cell subsets that are expanding in JIA patients is critical to decipher disease pathogenesis, and hyper-expanded T cells may represent novel therapeutic targets. Moreover, insight into the antigen specificity of local T cells may aid the discovery of disease-associated autoantigens.

Here, we had the unique opportunity to study autoimmune inflammation: (1) within different affected sites at one single time point (spatial dynamics), and (2) over time (temporal dynamics), to get a detailed understanding of T cell dynamics during human autoimmune inflammation. We profiled the T cell composition of inflammatory exudate as well as peripheral blood (PB) obtained from JIA patients using CyTOF. In addition, we performed TCRβ repertoire sequencing of Tregs and conventional CD4+ T cells (non-Tregs) derived from inflamed sites of JIA patients over time and space.

## Results

### Immune architecture of cellular infiltrates is similar between anatomically distinct inflamed sites

To study the peripheral and tissue-specific immune cell composition in autoimmune disease, we profiled peripheral blood mononuclear cells (PBMCs) and synovial fluid mononuclear cells (SFMCs) from JIA patients with both knees affected at the time of sampling using CyTOF (*Supplementary file 1*). T-distributed stochastic neighbor embedding (t-SNE) and k-means clustering identified 22 immune cell populations in the PB and synovial fluid (SF) compartments (*Figure 1A*, p<1e−21, *Figure 1— figure supplement 1A, B*). These populations could be broadly segregated into Treg (CD25+/FoxP3+),

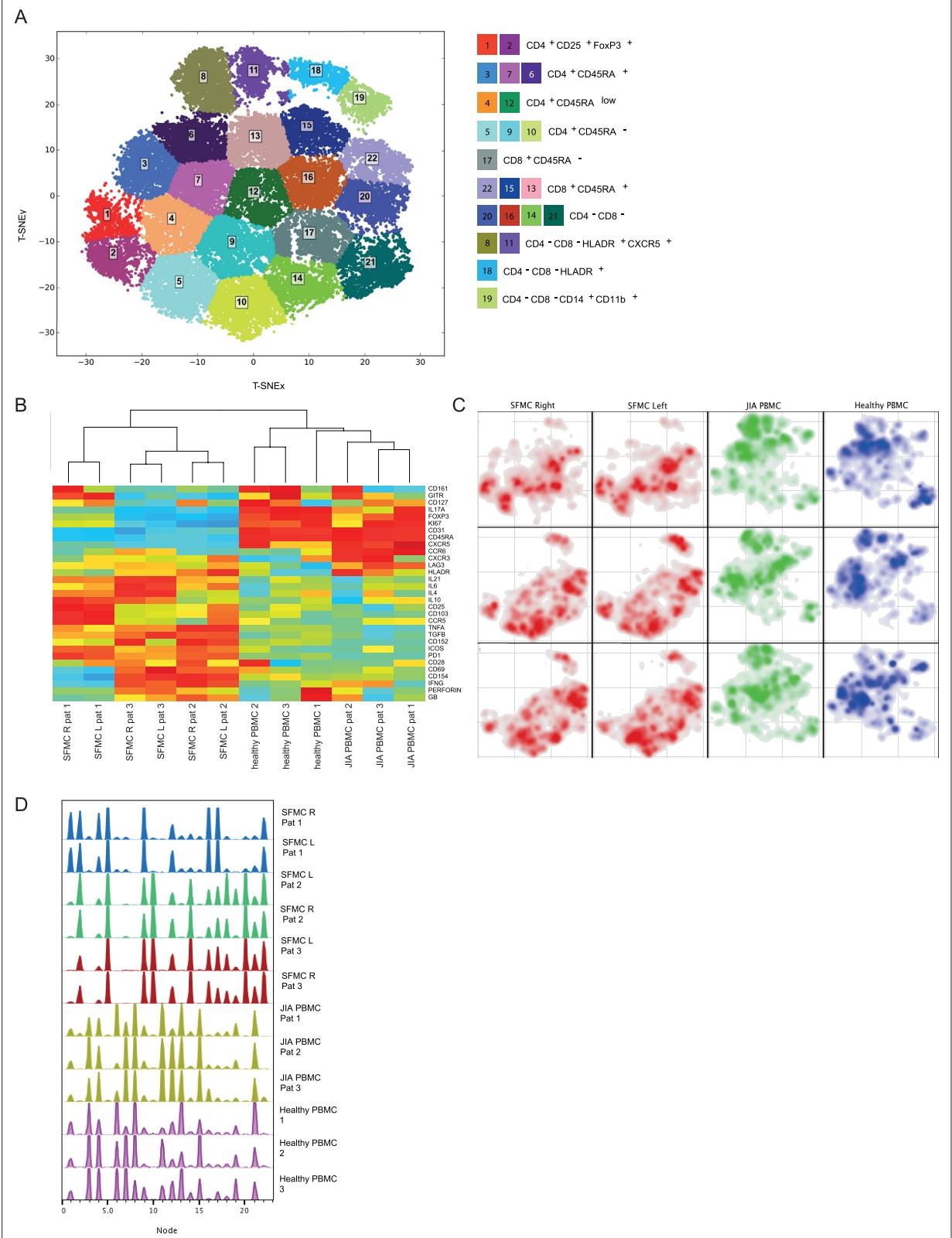

**Figure 1.** Overall immune architecture in left and right affected joints is very similar but distinct from peripheral blood. (**A**) Density maps based on t-SNE dimensional reduction and k-means clustering analysis on SF and PB samples, resulting in 22 cellular nodes. (**B**) Preliminary hierarchal clustering on the median expression of all markers, excluding lineage markers. (**C**) Density maps of immune cellular populations within the t-SNE maps. (**D**) Node

*Figure 1 continued on next page*

*Figure 1 continued*

frequency fingerprints showing the distribution across the nodes of SFMCs and PBMCs. PB, peripheral blood; PBMC, peripheral blood mononuclear cell; SF, synovial fluid; SFMC, synovial fluid mononuclear cell; t-SNE, t-distributed stochastic neighbor embedding.

The online version of this article includes the following figure supplement(s) for figure 1:

**Figure supplement 1.** Preliminary analysis reveals correlation between SFMC from distinct joints.

naive (CD45RA⁺), effector/memory (CD45RA⁻), and non-T cell populations (CD3⁻/CD4⁻/CD8⁻). Preliminary clustering of the median marker expression on T cells revealed a clear demarcation of SFMCs and PBMCs (*Figure 1B*), and a strong association of immune phenotypes between intra-individual paired knee SFMCs. Furthermore, density maps of immune cell populations within the t-SNE indicate strong dichotomy in the locations of SFMC and PBMC subsets (*Figure 1C*). Comparison of the node fingerprints between SFMC and PBMC samples (*Figure 1D*) revealed that SMFCs were enriched in CD4⁺CD25⁺FoxP3⁺ Tregs (node 2), and CD4⁺CD45RA⁻ memory T cells (nodes 5, 9, and 10), while PBMCs were enriched in CD45RA⁺ naive T cells (nodes 3, 6, 7, 13, and 15). Next to this, a strikingly similar cellular distribution profile was observed in the left and right knee joints of each JIA individual (*Figure 1C/D*). The correlation matrix of the entire spectrum of node frequencies demonstrated a strong positive correlation between the SFMCs and their left and right joints, and a strong negative correlation compared with the PBMC populations (*Figure 1—figure supplement 1C*). These results demonstrate that, while distinct differences in T cell signatures can be identified between PB and SF compartments, the phenotypic T cell architecture of distinct inflamed sites (left and right knees) are remarkably similar, indicating commonality in underlying disease etiology.

## Effector T cells and Tregs are phenotypically similar across distinct inflamed sites

Next, we functionally characterized SFMC-specific T cells, and found that CD4⁺ and CD8⁺ T cell subsets displayed an increased expression of pro-inflammatory cytokines (TNFα, IFNγ, and IL-6), indications of chronic TCR activation (PD1 and LAG3) (*Petrelli et al., 2018*) and a memory phenotype (CD45RA⁻), compared to their PBMC counterparts (*Figure 2—figure supplement 1A, B*, p<0.05). Remarkably, the cytokine diversity of CD4⁺ memory T cells revealed nearly identical profiles for the left and right knee joints for each individual (*Figure 2A*), with minor inter-individual differences. This trend in cytokine profile was also reflected in the CD8⁺CD45RA⁻ compartment (data not shown). The Treg (CD25⁺FOXP3⁺) population was significantly enriched in SFMC (*Figure 2B*, p<0.05, *Figure 2—figure supplement 1C, D*) with enhanced expression of memory (CD45RA⁻) and activation markers (HLA-DR/ICOS). Additionally, SFMC memory Tregs displayed a significantly higher proliferation (Ki67) as compared to SFMC effector memory T cells (*Figure 2B*, p<0.05), which was further confirmed by flow cytometry (*Figure 2—figure supplement 1E*). This indicates that Tregs belong to the most proliferative T cell subset in the inflamed environment. Moreover, memory Tregs showed very similar CTLA4/HLA-DR/ICOS/PD1 expression profiles in the left and right knee joints for each individual (*Figure 2C*). Altogether, these data demonstrate that within JIA patients, there is an identical T cell phenotypic and functional profile present at separate inflamed locations, with increased amounts of activated and proliferating Treg populations.

## Tregs are increased in autoimmune rheumatic disease and express markers of enhanced activation

To further study the relevance of Tregs in autoimmune rheumatic disease, we sought to validate our findings in data sets comparing SFMCs from rheumatoid arthritis (RA) to non-autoimmune osteoarthritis (OA, a degenerative, non-autoimmune disease of the joints, often used as a proxy for healthy donors). We obtained publicly available single-cell transcriptomics data (*Zhang et al., 2019*), and compared the gene expression profiles T cells from SFMC between RA and OA patients. We identified a cluster of CD4⁺FOXP3⁺ Tregs (characterized by high *CD4*, *FOXP3*, *PDCD1*, *CTLA4*, *TIGIT*, and *IL2RA* expression, *Figure 2—figure supplement 2A, B*) that showed increased frequency in RA patients compared to OA (*Figure 2—figure supplement 2C*). This increase of Tregs in RA SFMCs is consistent with the high frequency of Tregs that we observed in our JIA SFMC samples (*Figure 2B*). Additionally, the expression of markers related to chronic TCR activation (*PDCD1*, *CTLA4*, and *ICOS*), and cytokines

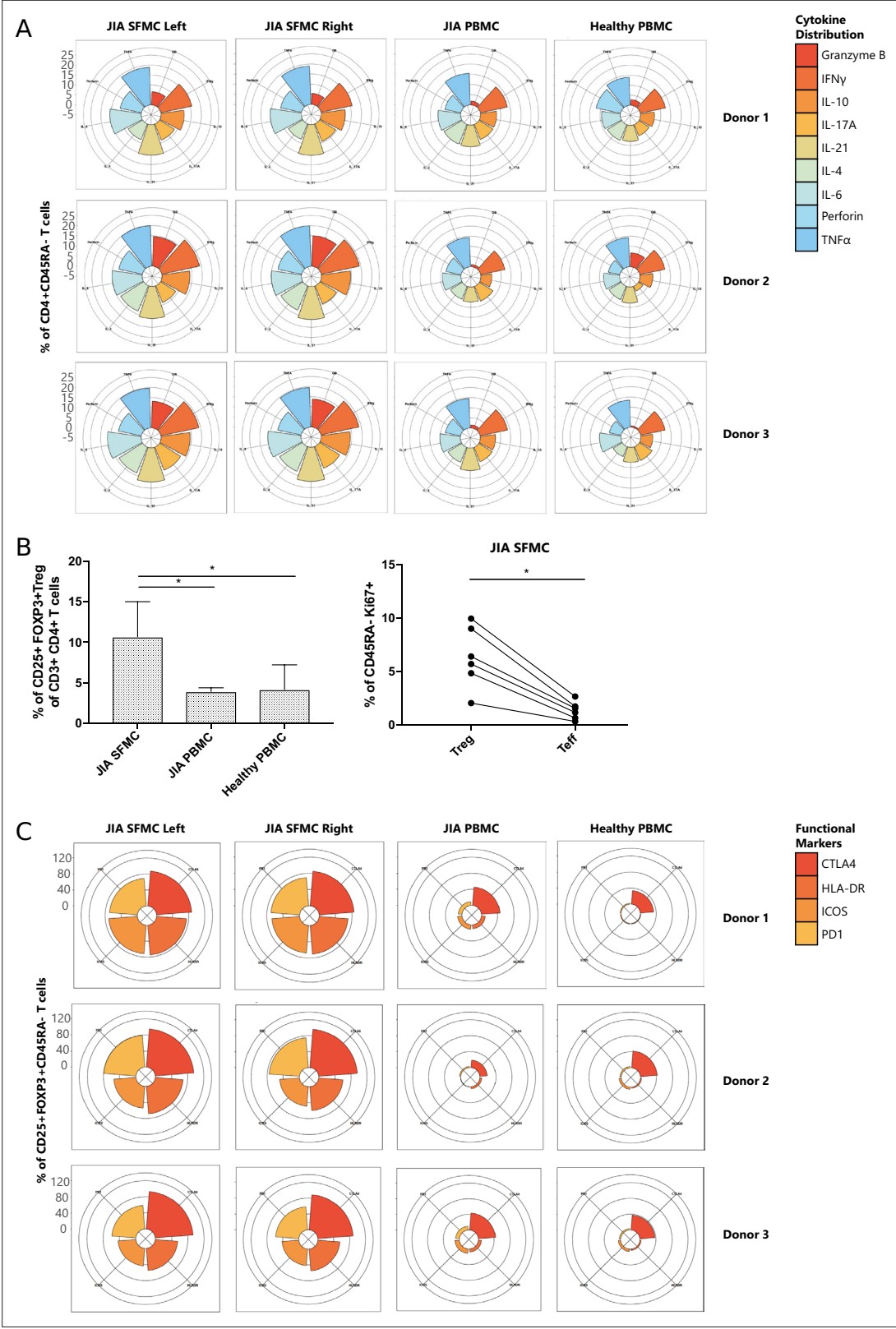

**Figure 2.** T cells display similar phenotypical and functional profiles at distinct inflamed locations. (**A**) Cytokine production of CD4⁺CD45RA⁻ memory T cells depicted in radar plots. Axis indicates the proportion of positive cells for individual cytokines (indicated by coloring) within the memory T cell fraction. (**B**) Percentage CD25⁺FOXP3⁺ Treg of CD3⁺CD4⁺ cells in SFMC and PBMC of JIA patients and healthy children, and percentage of Ki67⁺ cells within CD45RA⁻ cells in Treg and non-Treg in SFMC (nonparametric Mann-Whitney, *=p<0.05). For SFMCs, data from the right and left knee joints for

*Figure 2 continued on next page*

*Figure 2 continued*

all patients is shown. (**C**) Expression of functional markers by CD25+FOXP3+CD45RA− cells. JIA, juvenile idiopathic arthritis; PBMC, peripheral blood mononuclear cell; SFMC, synovial fluid mononuclear cell.

The online version of this article includes the following figure supplement(s) for figure 2:

**Figure supplement 1.** JIA SFMCs display an activated expression profile.

**Figure supplement 2.** Tregs are increased in autoimmune rheumatic disease and express markers of enhanced activation.

(*TNF*, *IFNG*, and *GZMB*) were significantly increased in RA compared to OA (*Figure 2—figure supplement 2D*), in line with what we observed in JIA SFMC (*Figure 2A/B*).

## Hyper-expanded T cell clones are shared between left and right joints

To study whether the same expanded T cell clones infiltrate multiple joints, we performed TCR sequencing for similar numbers of CD3+CD4+CD25+CD127$^{low}$ Tregs and CD3+CD4+CD25−CD127+ non-Tregs sorted from affected joints of JIA patients, derived from the same donors and time points as the ones used for CyTOF analysis regarding the first two patients. Within the inflamed joints, clonally expanded cells were detected, which was more pronounced for Tregs than non-Tregs (*Figure 3A*). In line with the CyTOF analysis, the distribution of T cell clones was highly similar between left and right joints, both for Tregs and non-Tregs. Hyper-expanded T cells were further studied by the sequential intersection of the most abundant TCRβ clonotypes across samples. We found a high degree of sharing between two affected joints, while a small fraction of clones was shared between SF and PB (*Figure 3B*). Moreover, sharing of clones between two joints was more evident for Tregs than non-Tregs (*Figure 3B*).

Detailed analysis further revealed that frequencies of hyper-expanded T cells were highly conserved between distinct anatomical sites, with the most dominant clones also detectable in PB (*Figure 3C*). To assess whether dominant clones were shared as a result of high generation probability (p$_{gen}$, convergent recombination; *Sethna et al., 2019*), or in response to antigen (convergent selection), we calculated the p$_{gens}$ of shared and non-shared clones and correlated these with their respective frequencies. Frequencies of shared clones were not correlated with p$_{gen}$ (*Figure 3D*), while frequencies of non-shared clones showed a significant positive correlation with p$_{gen}$ (*Figure 3E*). Notably, this correlation was more pronounced for non-Tregs (*Figure 3E*), indicating either bystander activation or non-antigen-specific circulation of the non-shared TCR clones in the non-Treg compartment. In summary, both non-Treg and Treg hyper-expanded T cell clones are shared between inflamed joints. This overlap is most pronounced for Treg, with the highly dominant Treg clones in SF also being detectable in circulation, likely driven by responses to shared antigens.

To investigate whether the TCR repertoire of JIA patients differs from healthy, we compared the repertoire diversity of PB JIA Tregs to the diversity of Tregs obtained from PB of healthy individuals from a publicly available data set (*Kasatskaya et al., 2020*). Sample-based rarefaction analysis showed that the estimated species richness and Shannon diversity was significantly lower for JIA Tregs compared to Tregs obtained from healthy donors (*Figure 3—figure supplement 1*), demonstrating that the JIA Treg repertoire is less diverse than healthy.

## Dominant clones persist over time during relapsing-remitting disease

Next, to study the temporal dynamics of T cells in JIA, we profiled the Treg and non-Treg TCRβ repertoire of SF and PB samples from five JIA patients over time (*Figure 4A*, *Figure 4—figure supplement 1*). Repertoire overlap analysis showed that TCRβs of SF Tregs were highly shared within patients over time (*Figure 4A*), which was also conserved across different joints (*Figure 4A/B*, *Figure 4—figure supplement 2A*). In contrast, TCRβs from PB did not cluster together over time, and showed much less overlap with their synovial counterparts (*Figure 4A*). More detailed analysis showed that frequencies of shared TCRβs were also consistent over time, with the most dominant T cell clones having the highest degree of sharing (*Figure 4C*, *Figure 4—figure supplement 3*). Again, this phenomenon was more pronounced in Tregs from SF compared to PB (*Figure 4C*), although the most dominant clones from SF were also detectable in PB (*Figure 4—figure supplement 4*). Moreover, persistent TCRβs with high abundance were not driven by recombination bias (*Figure 4D*), similar to what was observed for T cell clones shared between two knees sampled at the same time point (*Figure 3D*).

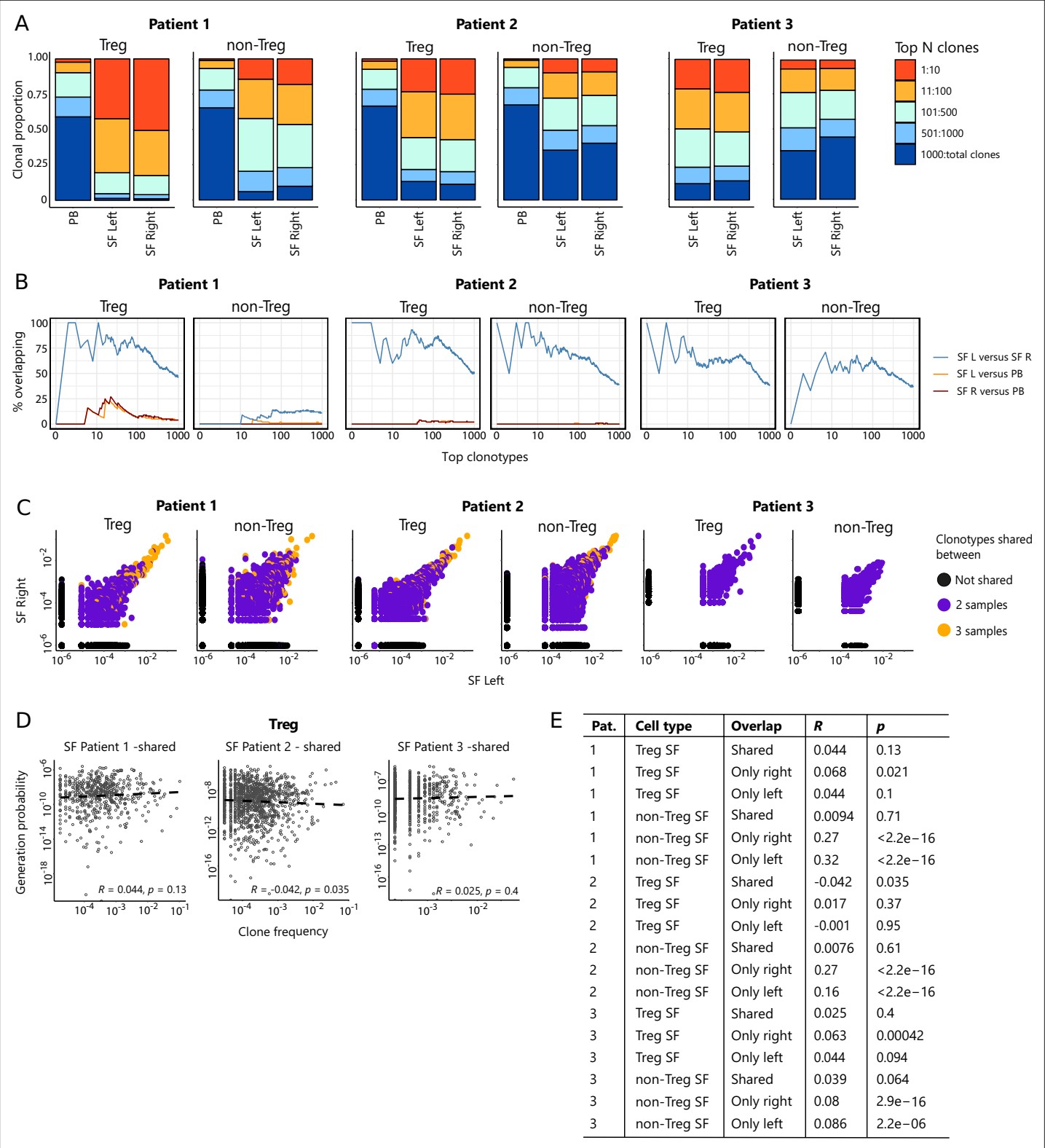

**Figure 3.** Highly dominant T cell clones are shared in synovial fluid (SF) from left and right joints and peripheral blood (PB). (**A**) Clonal proportions of the TCRβ clones as detected in Treg and non-Treg sorted from PBMC, SF left joint, SF right joint of two different JIA patients. (**B**) Sequential intersection of abundant TCRβ clonotypes (based on amino acid sequence) across samples. Top clonotypes (ranging from 1 to 1000) are given on the x-axis, with the percentage of sequences overlapping between two given samples on the y-axis. For patient 3, no PB sample was available. (**C**) Frequency plots showing the overlapping Treg and non-Treg clones between left joint derived SF (x-axis) and right joint derived SF (y-axis), with color coding highlighting the

*Figure 3 continued on next page*

*Figure 3 continued*

clones that are shared with none of the other samples (black circle), shared in two samples (purple) and all three samples (PB, SF left, SF right; yellow). (**D**) Correlation (linear regression, dashed line) between frequency (x-axis) and generation probability (y-axis) of TCR clones shared across SF two samples. (**E**) Results of correlation between frequency and generation probability across all samples. p, p value; Pat., patient; PBMC, peripheral blood mononuclear cell; R, Spearman's Rho; SF, synovial fluid.

The online version of this article includes the following figure supplement(s) for figure 3:

**Figure supplement 1.** The JIA peripheral Treg repertoire is less diverse than healthy.

Next, we repeated our analysis on TCRβ sequences of non-Tregs from the same samples. Although non-Tregs also display sharing of TCRβ sequences over time (*Figure 5A/B*, *Figure 4—figure supplement 2B*), the degree of sharing was less pronounced compared to Tregs (*Figure 4A*). Frequencies of highly shared TCRβs in non-Tregs were also consistent over time (*Figure 5C*, *Figure 5—figure supplement 1*), and not driven by recombination bias (*Figure 5D*; *Figure 5—figure supplement 1*). Collectively, these data show that during relapsing-remitting disease, persistent dominant T cell clones are taking part in the local immune response in JIA patients, and this phenomenon is more pronounced for Tregs than non-Tregs.

## Patterns in similar TCR sequences are shared between JIA patient knees

Recent studies have demonstrated that immune responses against a particular antigen involve T cell clones with similar TCR sequences (*Dash et al., 2017*; *Glanville et al., 2017*; *Pogorelyy et al., 2019*). To investigate whether persistent T cell clones in JIA cluster together with other, similar T cell clones involved in responses against the same antigens, we performed TCR similarity analysis, focusing on SF samples obtained from two affected knees. We constructed similarity networks for JIA patients and compared these to networks generated from random repertoires with the same number of TCRβ sequences (*Figure 6A*). TCR networks from JIA patients were highly connected (more than expected by chance), showing that patient repertoires exhibit a high degree of sequence similarity (*Figure 6B*). Moreover, in the random repertoires, clusters were less mixed (indicated by a high cluster purity) than JIA networks (*Figure 6C*), highlighting that TCRs from JIA samples display higher sequence similarity than expected by chance. Overall, these results show that the SF Treg repertoire is highly skewed by antigenic selection.

## Discussion

In this study, we provide the first CyTOF and TCRβ sequencing analysis of purified Tregs and non-Tregs, uncovering their spatial and temporal behavior in a human autoimmune disease setting. Although the antigen(s) driving T cell activation and expansion in JIA remain elusive, our data provide strong support for the presence of ubiquitously expressed autoantigens given the observed overlap in dominant clones over time and in space. Given the tissue restrictive character of the JIA, it is tempting to speculate that the potential antigen would be joint-specific, although it has been shown that ubiquitously expressed autoantigens can also induce joint-specific autoimmune disease (*Ito et al., 2014*; *Mandik-Nayak et al., 2002*). We show that SF Tregs have high expression of Ki67 (marking proliferation and thus recent antigen encounter), suggesting that these cells actively respond to synovial antigens. Moreover, we show that the expansion of dominant TCR clones is not dependent on generation probabilities, further highlighting that antigens are driving T cell activation. Further support for the hypothesis that persistent, hyper-expanded Tregs found in JIA SF are auto-reactive is provided by a recent study performed in mice with type 1 diabetes, where Tregs with a high degree of self-reactivity were found to be expanding locally in affected pancreatic islets and displayed a specific profile with elevated levels of GITR, CTLA-4, ICOS, and Ki67, very similar to our observations (*Sprouse et al., 2018*). Moreover, in single-cell transcriptomics data (*Zhang et al., 2019*), we identified a cluster of CD4$^+$FOXP3$^+$ Tregs that showed increased frequency in SF from RA patients, along with increased cytokine expression and expression of markers of chronic TCR activation. These findings match previous transcriptomic analyses, which showed that Tregs from SF of RA patients display a very similar gene expression signature as Tregs from JIA patients (*Mijnheer et al., 2021*). These results further highlight that Tregs in rheumatic autoimmune disease are by characterized

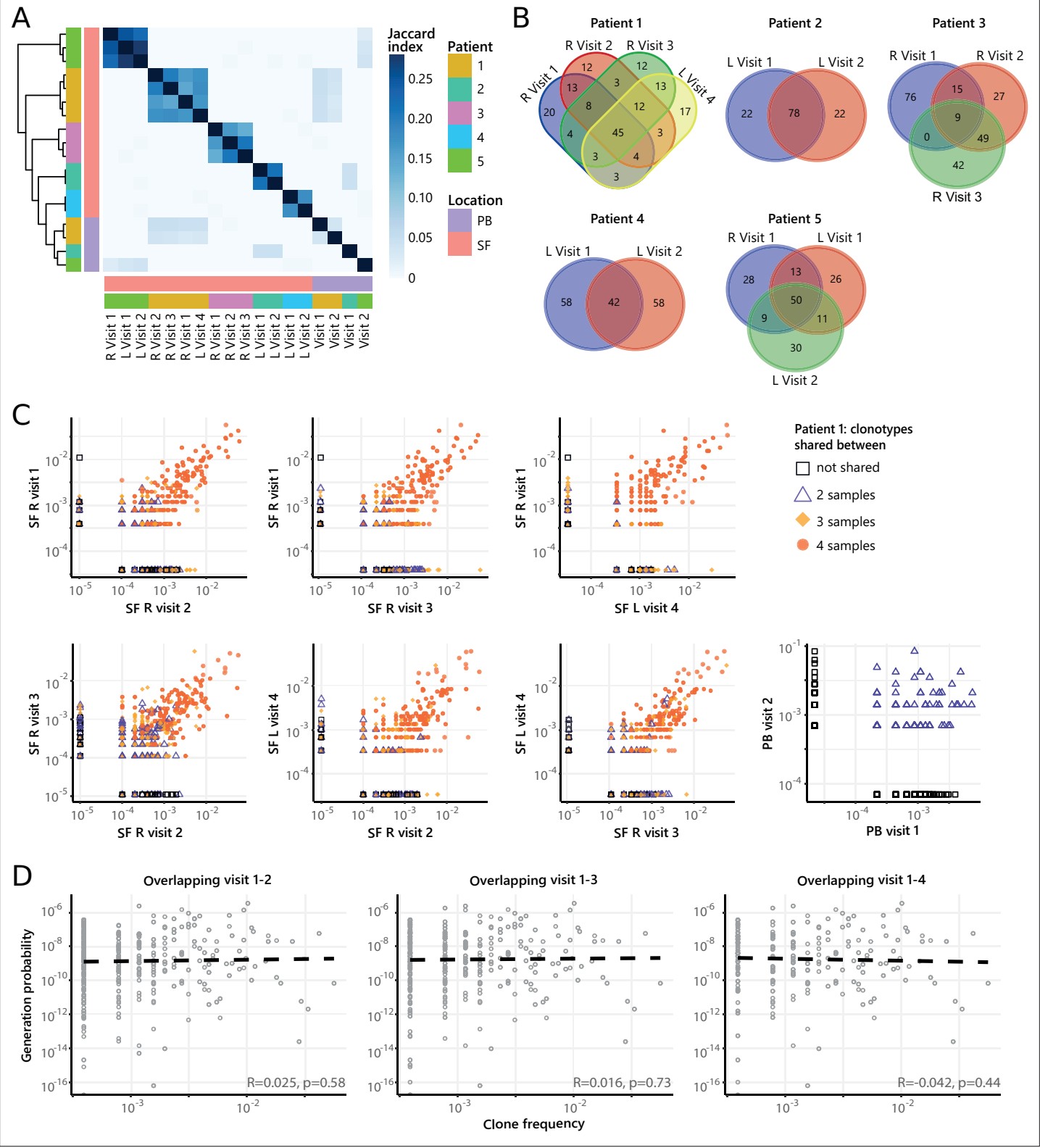

**Figure 4.** Persistence of Treg clones over the course of relapse remitting disease. (**A**) Heatmap showing overlap (Jaccard index, light blue = limited overlap, darkblue = high overlap) of Treg derived TCRβ sequences obtained from SF or PB from JIA patients over time. L=left knee, R=right knee. (**B**) Venn diagrams displaying the 100 most abundant unique TCRβ clones, defined by amino acid sequence, for longitudinal SF samples from all patients. (**C**) Frequency plots showing the overlapping Treg clones between visits for SF and PB, with color coding and shapes highlighting the number

*Figure 4 continued on next page*

Figure 4 continued

of samples in which unique clones are found. L=left; R=right. (**D**) Correlation (linear regression, dashed line) between frequency (x-axis) and generation probability (y-axis) of TCR clones shared across two visits for SF samples. PB, peripheral blood; SF, synovial fluid; TCR, T cell receptor.

The online version of this article includes the following figure supplement(s) for figure 4:

**Figure supplement 1.** Longitudinal sampling timelines of JIA patients.

**Figure supplement 2.** TCR overlap analysis.

**Figure supplement 3.** JIA Treg TCR frequencies over time in the remaining four patients.

**Figure supplement 4.** Frequencies of TCRs from persistent Tregs shared across SF and PB samples.

inflammation-related molecular signatures potentially relevant for autoimmune disease. Further validation of our observations in larger cohorts of JIA patients should help to substantiate these results and aid the identification of pathogenic Treg populations across patients.

Our data demonstrated that dominant T cell clones in SF can be traced back in circulation. Together with observations that similar T cell clones are detected in multiple affected joints and the obvious overlap in immune cell composition, this strongly suggests that T cells migrate from the joint to PB and vice versa. This could mean that Tregs are either recirculating, or actively being replenished from circulating (precursor) T cells. These observations are in line with other recent studies in arthritis showing that synovial CD4$^+$T cells and Treg clones can also be detected in PB (**Rossetti et al., 2017**; **Spreafico et al., 2016**), where their presence correlates with disease activity and response to therapy (**Rossetti et al., 2017**; **Lutter et al., 2018**). Moreover, for refractory systemic JIA patients who underwent autologous hematopoietic stem cell transplantation (aHSCT), transplant outcome was shown to be dependent upon the diversity of circulating Tregs (**Delemarre et al., 2016**; **Lutter et al., 2018**). This knowledge, combined with our findings that the same T cell clones dominate the immune response at different sites of inflammation and the persistence of the same clones in the relapsing-remitting course of disease, strengthen the possibility to use circulating disease-associated T cell clones for disease monitoring or prognostic purposes. Additionally, we also found the repertoire diversity of circulating Tregs from JIA patients to be decreased compared to healthy. This is also in line with previous findings demonstrating that the diversity of the Treg TCR repertoire in systemic JIA patients is reduced, especially in treatment refractory patients, and increases after aHSCT (**Delemarre et al., 2016**). However, to accurately monitor and predict which T cell clones from PB are implicated in active immune processes in joints, more detailed phenotyping is needed to fully characterize the functional profile and origins of dominant clones. Multi-omic single-cell profiling to link TCR specificity with gene expression will help to bring this closer to the clinic.

The existence of a temporal and spatially persistent clonal Treg TCR repertoire, raises the question to what degree clonally expanded Tregs can modulate inflammation over the course of an autoimmune response. Various studies have shown that Tregs in JIA maintain their suppressive capacity, but local effector T cells are resistant to this suppression (**Wehrens et al., 2011**; **Haufe et al., 2011**). Thus, the clonotypic expansion in SF Treg cells might reflect an insufficient attempt to control expanding effector T cells. The importance of a diverse Treg repertoire is shown in several mouse models (**Adeegbe et al., 2010**; **Föhse et al., 2011**; **Yu et al., 2017**; **Nishio et al., 2015**). **Föhse et al., 2011** showed that Tregs with a higher diversity are able to expand more efficiently compared to Treg with a lower diversity in mice with TCR restricted conventional T cells. It has been suggested that this is due to the TCR diverse Tregs having access to more ligands and as a result being able to out-compete the TCR-restricted Treg cells (**Wing and Sakaguchi, 2011**). However, this applies for circulating Treg, and whether this would also be important for Treg in tissues is not known. The finding that tissue Treg residing in healthy tissues also show a considerable oligoclonality regarding their TCR repertoire may indicate that this is a normal feature (**Burzyn et al., 2013**; **Sanchez Rodriguez et al., 2014**). Additionally, it was recently shown that a diverse Treg repertoire in mice is especially needed to control Th1 responses, whereas Th2 and Th17 responses were still suppressed by single Treg clones (**Levine et al., 2017**). This could be an explanation why the Th1-rich SF environment is poorly controlled by the large amount of clonally expanded Tregs. Thus, hyper-expanded Tregs alone might not be sufficient to prevent or inhibit autoimmune responses, and future Treg-centric therapies should take this into account.

In this study, we sequenced the β-chain of the TCR and not the α-chain. The identified dominant TCRβ clones can pair with several α-chains, possibly leading to less overlapping TCR repertoire and a different Ag specificity. Future sequencing of both TCR chains will provide insight into the total TCR repertoire. Next to that, we are aware of a possible amplification bias because of a difference in efficiency of PCR primers. However, in our analysis approach, we attempted to control as much as possible for such biases. An interesting next step would be to combine single-cell RNA-sequencing with identification of the TCR to directly link the expression profile of a given cell to its TCR clonotype and facilitate the identification of the antigenic target and its HLA class II restriction.

In conclusion, we show that in SF the immune cell architecture is marked by inflammatory responses of activated effector T cells as well as activated and highly expanding Tregs. The remarkable overlap in immune cell composition as well as the dominant clones over time and in space provide indications for a powerful driving force that shapes the local T cell response during joint inflammation. The presence of these inflammation-associated clones in the circulation provide promising perspectives for use in disease monitoring. Moreover, the high degree of sequence similarity observed between Treg clones obtained from distinct inflamed joints indicates that antigen selection significantly reshapes the local Treg repertoire. Further research is needed to pinpoint these driving antigens and create opportunities to target disease-specific T cells.

# Materials and methods

## Key resources table

| Reagent type (species) or resource | Designation | Source or reference | Identifiers | Additional information |
|---|---|---|---|---|
| Biological sample (human) | Peripheral blood | University Medical Center Utrecht | | Healthy donors and JIA patients |
| Biological sample (human) | Synovial fluid | University Medical Center Utrecht | | JIA patients |
| Antibody | Anti-human CD3 (UCHT1) (Mouse monoclonal) | BioLegend | Cat#: 300402 | CyTOF (5µg/ml) |
| Antibody | Anti-human CD4 (SK3) (Mouse monoclonal) | BioLegend | Cat#: 344625 | CyTOF (5µg/ml) |
| Antibody | Anti-human CD8 (SK1) (Mouse monoclonal) | BioLegend | Cat#: 344727 | CyTOF (5µg/ml) |
| Antibody | Anti-human CD11b (ICRF44) (Mouse monoclonal) | BioLegend | Cat#: 301302 | CyTOF (5µg/ml) |
| Antibody | Anti-human CD16 (3G8) (Mouse monoclonal) | Fluidigm | Cat#: 3209002B | CyTOF (5µg/ml) |
| Antibody | Anti-human CD14 (M5E2) (Mouse monoclonal) | BioLegend | Cat#: 301843 | CyTOF (5µg/ml) |
| Antibody | Anti-human IL-4 (8D4-8) (Mouse monoclonal) | BioLegend | Cat#: 500707 | CyTOF (5µg/ml) |
| Antibody | Anti-human IFN-g (B27) (Mouse monoclonal) | BioLegend | Cat#: 506513 | CyTOF (5µg/ml) |
| Antibody | Anti-human IL-17A (BL168) (Mouse monoclonal) | BioLegend | Cat#: 512302 | CyTOF (5µg/ml) |
| Antibody | Anti-human IL-21 (3A4-N2) (Mouse monoclonal) | BioLegend | Cat#: 513009 | CyTOF (5µg/ml) |
| Antibody | Anti-human CD161 (HP-3G10) (Mouse monoclonal) | BioLegend | Cat#: 339902 | CyTOF (5µg/ml) |
| Antibody | Anti-human CD45RA (HI100) (Mouse monoclonal) | BioLegend | Cat#: 304102 | CyTOF (5µg/ml) |
| Antibody | Anti-human CD69 (FN50) (Mouse monoclonal) | BioLegend | Cat#: 310902 | CyTOF (5µg/ml) |

*Continued on next page*

*Continued*

| Reagent type (species) or resource | Designation | Source or reference | Identifiers | Additional information |
|---|---|---|---|---|
| Antibody | Anti-human CD28 (CD28.2) (Mouse monoclonal) | BioLegend | Cat#: 302923 | CyTOF (5µg/ml) |
| Antibody | Anti-human CD152 (BNI3) (Mouse monoclonal) | BioLegend | Cat#: 555851 | CyTOF (5µg/ml) |
| Antibody | Anti-human CD154 (24-31) (Mouse monoclonal) | BioLegend | Cat#: 310835 | CyTOF (5µg/ml) |
| Antibody | Anti-human HLA-DR (L243) (Mouse monoclonal) | BioLegend | Cat#: 307612 | CyTOF (5µg/ml) |
| Antibody | Anti-human LAG3 (17B4) (Mouse monoclonal) | Abcam | Cat#: ab40466 | CyTOF (5µg/ml) |
| Antibody | Anti-human PD1 (EH12.2H7) (Mouse monoclonal) | BioLegend | Cat#: 329941 | CyTOF (5µg/ml) |
| Antibody | Anti-human Ki67 (20Raj1) (Mouse monoclonal) | Thermo Fisher Scientific/eBioscience | Cat#: 14-5699-82 | CyTOF (5µg/ml) |
| Antibody | Anti-human ICOS (C398.4A) (Armenian Hamster monoclonal) | BioLegend | Cat#: 313512 | CyTOF (5µg/ml) |
| Antibody | Anti-human CD31 (WM59) (Mouse monoclonal) | BioLegend | Cat#: 303102 | CyTOF (5µg/ml) |
| Antibody | Anti-human CD103 (B-Ly7) (Mouse monoclonal) | Thermo Fisher Scientific/eBioscience | Cat#: 14-1038-82 | CyTOF (5µg/ml) |
| Antibody | Anti-human CXCR3 (G025H7) (Mouse monoclonal) | BioLegend | Cat#: 353718 | CyTOF (5µg/ml) |
| Antibody | Anti-human CXCR5 (RF8B2) (Rat monoclonal) | BD Biosciences | Cat#: 552032 | CyTOF (5µg/ml) |
| Antibody | Anti-human CCR5 (NP-6G4) (Mouse monoclonal) | Abcam | Cat#: ab115738 | CyTOF (5µg/ml) |
| Antibody | Anti-human CCR6 (G034E3) (Mouse monoclonal) | BioLegend | Cat#: 353402 | CyTOF (5µg/ml) |
| Antibody | Anti-human CD25 (M-A251) (Mouse monoclonal) | BD Biosciences | Cat#: 555429 | CyTOF (5µg/ml) |
| Antibody | Anti-human CD127 (A019D5) (Mouse monoclonal) | BioLegend | Cat#: 351302 | CyTOF (5µg/ml) |
| Antibody | Anti-human FOXP3 (PCH101) (Rat monoclonal) | Thermo Fisher Scientific/eBioscience | Cat#: 14-4776-82 | CyTOF (5µg/ml) |
| Antibody | Anti-human GITR (621) (Mouse monoclonal) | BioLegend | Cat#: 311602 | CyTOF (5µg/ml) |
| Antibody | Anti-human TGF-B (TW4-2F8) (Mouse monoclonal) | BioLegend | Cat#: 349602 | CyTOF (5µg/ml) |
| Antibody | Anti-human IL-10 (JES3-9D7) (Rat monoclonal) | BioLegend | Cat#: 501402 | CyTOF (5µg/ml) |
| Antibody | Anti-human TNF-alpha (Mab11) (Mouse monoclonal) | BioLegend | Cat#: 502902 | CyTOF (5µg/ml) |
| Antibody | Anti-human IL-6 (MQ2-13A5) (Rat monoclonal) | Thermo Fisher Scientific/eBioscience | Cat#: 16-7069-85 | CyTOF (5µg/ml) |
| Antibody | Anti-human Granzyme B (CLB-GB11) (Mouse monoclonal) | Abcam | Cat#: ab103159 | CyTOF (5µg/ml) |
| Antibody | Anti-human Perforin (B-D48) (Mouse monoclonal) | Abcam | Cat#: ab47225 | CyTOF (5µg/ml) |

*Continued on next page*

*Continued*

| Reagent type (species) or resource | Designation | Source or reference | Identifiers | Additional information |
|---|---|---|---|---|
| Antibody | Anti-human CD45-A (HI30) (Mouse monoclonal) | Fluidigm | Cat#: 3089003B | CyTOF (5µg/ml) |
| Antibody | Anti-human CD45-B, C or D (HI30) (Mouse monoclonal) | BioLegend | Cat#: 304002 | CyTOF (5µg/ml) |
| Antibody | DNA (singlets) Cell-ID Intercalator-Ir | Fluidigm | Cat#: 201192 | CyTOF (5µg/ml) |
| Antibody | Cisplatin (Live/Dead) | Sigma-Aldrich | Cat#: 479306-1G | CyTOF (5µg/ml) |
| Antibody | Anti-human CD3-BV510 (UCHT1) (Mouse monoclonal) | BioLegend | Cat#: 300448 | FACS (dilution 1:50) |
| Antibody | Anti-human CD4-FITC (SK3) (Mouse monoclonal) | eBioscience | Cat#: 11-0047-42 | FACS (dilution 1:100) |
| Antibody | Anti-human CD25-PE/Cy7 (2A3) (Mouse monoclonal) | BD | Cat#: 335789 | FACS (dilution 1:50) |
| Antibody | Anti-human CD127-AF647 (A019D5) (Mouse monolonal) | BioLegend | Cat#: 351318 | FACS (dilution 1:50) |
| Antibody | Anti-human FOXP3-eF450 (PCH101) (Rat monoclonal) | eBioscience | Cat#: 48-4776-42 | FACS (dilution 1:50) |
| Chemical compound, drug | Phorbol 12-myristate 13-acetate (PMA) | Sigma-Aldrich | Cat#: P1585 | |
| Chemical compound, drug | Ionomycin | Sigma-Aldrich | Cat#: I9657 | |
| Chemical compound, drug | Brefeldin A | eBioscience | Cat#: 00-4506-51 | |
| Chemical compound, drug | Monensin | BioLegend | Cat#: 420701 | |
| Chemical compound, drug | Intracellular Fixation & Permeabilization Buffer Set | eBioscience | Cat#: 88-8824-00 | |
| Chemical compound, drug | EQ Four Element Calibration beads | Fluidigm | Cat#: NC1307119 | |
| Commercial assay or kit | RNeasy mini kit | Qiagen | Cat#: 74104 | |
| Commercial assay or kit | RNeasy Micro Kit | Qiagen | Cat#: 74004 | |
| Commercial assay or kit | SMARTer RACE cDNA Amplification kit | Clontech | Cat#: 634923 | |
| Commercial assay or kit | NGSgo-LibrX | GenDx | Cat#: 2342605 | |
| Commercial assay or kit | NGSgo-IndX | GenDx | Cat#: 2342153 | |
| Software, algorithm | FlowJo (v.10.2) | TreeStar | | |
| Software, algorithm | MarVis | https://doi.org/10.1186/1471-2105-10-92 | | |
| Software, algorithm | Seurat (v.4.1.1) | Massachusetts Institute of Technology (MIT) | | |
| Software, algorithm | iNEXT (v.3.0.0) | https://doi.org/10.1111/2041-210X.12613 | | |
| Software, algorithm | GraphPad Prism (v.7.0) | GraphPad | | |

*Continued on next page*

*Continued*

| Reagent type (species) or resource | Designation | Source or reference | Identifiers | Additional information |
|---|---|---|---|---|
| Software, algorithm | RTCR | https://doi.org/10.1093/bioinformatics/btw339 | | |
| Software, algorithm | OLGA | https://doi.org/10.1093/bioinformatics/btz035 | | |

## Collection of SF and PB samples

Patients with JIA were enrolled at the University Medical Center Utrecht (The Netherlands) (*Supplementary file 1*). A total of nine JIA patients were included in this study. Of these, n=2 were diagnosed with extended oligo JIA, n=2 with rheumatoid factor negative poly-articular JIA, and n=5 with oligo JIA, according to the revised criteria for JIA (*Petty et al., 1998*). The average age at the time of inclusion was 13.1 years (range 3.2–18.1 years) with a disease duration of 7.3 years (range 0.4–14.2 years). Patients included for CyTOF and TCR sequencing analysis of two affected knee joints were all treated with non-steroidal anti-inflammatory drugs (NSAIDs) or methotrexate, but no biologicals at the time of sampling. For sequential TCR sequencing analysis, we included patients with a relapsing disease course. At the first time point, all patients were treated with NSAIDs or methotrexate, but no biologicals. During the follow-up (after experiencing a relapse of disease), patients were treated with disease-modifying anti-rheumatic drugs (leflunomide) and/or biologicals (humira).

PB of JIA patients was obtained via veni-puncture or intravenous drip, while SF was obtained by therapeutic joint aspiration of affected joints. Informed consent was obtained from all patients either directly or from parents/guardians when the patients were younger than 12 years of age. The study was conducted in accordance with the Institutional Review Board of the University Medical Center Utrecht (approval no. 11-499/C), in compliance with the Declaration of Helsinki. PB from n=3 healthy children (average age 15.1 years with range 14.7–15.4 years) was obtained from a cohort of control subjects for a case-control clinical study (*Supplementary file 1*).

## Cell isolation

For cell isolation, SF was incubated with hyaluronidase (Sigma-Aldrich, St. Louis, MO) for 30 min at 37°C to break down hyaluronic acid. SFMCs and PBMCs were isolated using Ficoll Isopaque density gradient centrifugation (GE Healthcare Bio-Sciences AB, Uppsala, Sweden), and were used after freezing in fetal calf serum (FCS) (Invitrogen, Waltham, MA) containing 10% DMSO (Sigma-Aldrich).

## Flow cytometry and cell sorting

For TCR sequencing purposes, CD3$^+$CD4$^+$CD25$^{high}$CD127$^{low}$ Tregs and CD3$^+$CD4+CD25$^{low/int}$CD127$^{int/high}$ non-Tregs were isolated from frozen PBMCs and SFMCs, using the FACS Aria III (BD, Franklin Lakes, NJ). Antibodies used for sorting were: anti-human CD3-BV510 (BioLegend, San Diego, CA), CD4-FITC (eBioscience, Frankfurt am Main, Germany), CD25-PE/Cy7 (BD), and CD127-AF647 (BioLegend). To check for FOXP3 expression of the sorted populations anti-human FOXP3-eF450 (eBioscience) was used.

## CyTOF and CyTOF data analysis

Frozen PBMCs and SFMCs were thawed and stained with a T cell focused panel of 37 heavy metal-conjugated antibodies (*Supplementary file 2*), as previously described (*Chew et al., 2019*), and analyzed by CyTOF-Helios (Fluidigm, San Francisco, CA). Briefly, PBMCs were stimulated with or without phorbol 12-myristate 13-acetate (150 ng/ml, Sigma-Aldrich) and ionomycin (750 ng/ml, Sigma-Aldrich) for 4 h, and blocked with secretory inhibitors, brefeldin A (1:1000, eBioscience) and monensin (1:1000, BioLegend) for the last 2 h. The cells were then washed and stained with cell viability dye cisplatin (200 μM, Sigma-Aldrich). Each individual sample was barcoded with a unique combination of anti-CD45 conjugated with either heavy metal 89, 115, 141, or 167, as previously described (*Lai et al., 2015*). Barcoded cells were washed and stained with the surface antibody cocktail for 30 min on ice, and subsequently washed and re-suspended in fixation/permeabilization buffer (permeabilization buffer, eBioscience) for 45 min on ice. Permeabilized cells were subsequently stained with an intra-cellular antibody cocktail for 45 min on ice, followed by staining with a DNA intercalator Ir-191/193

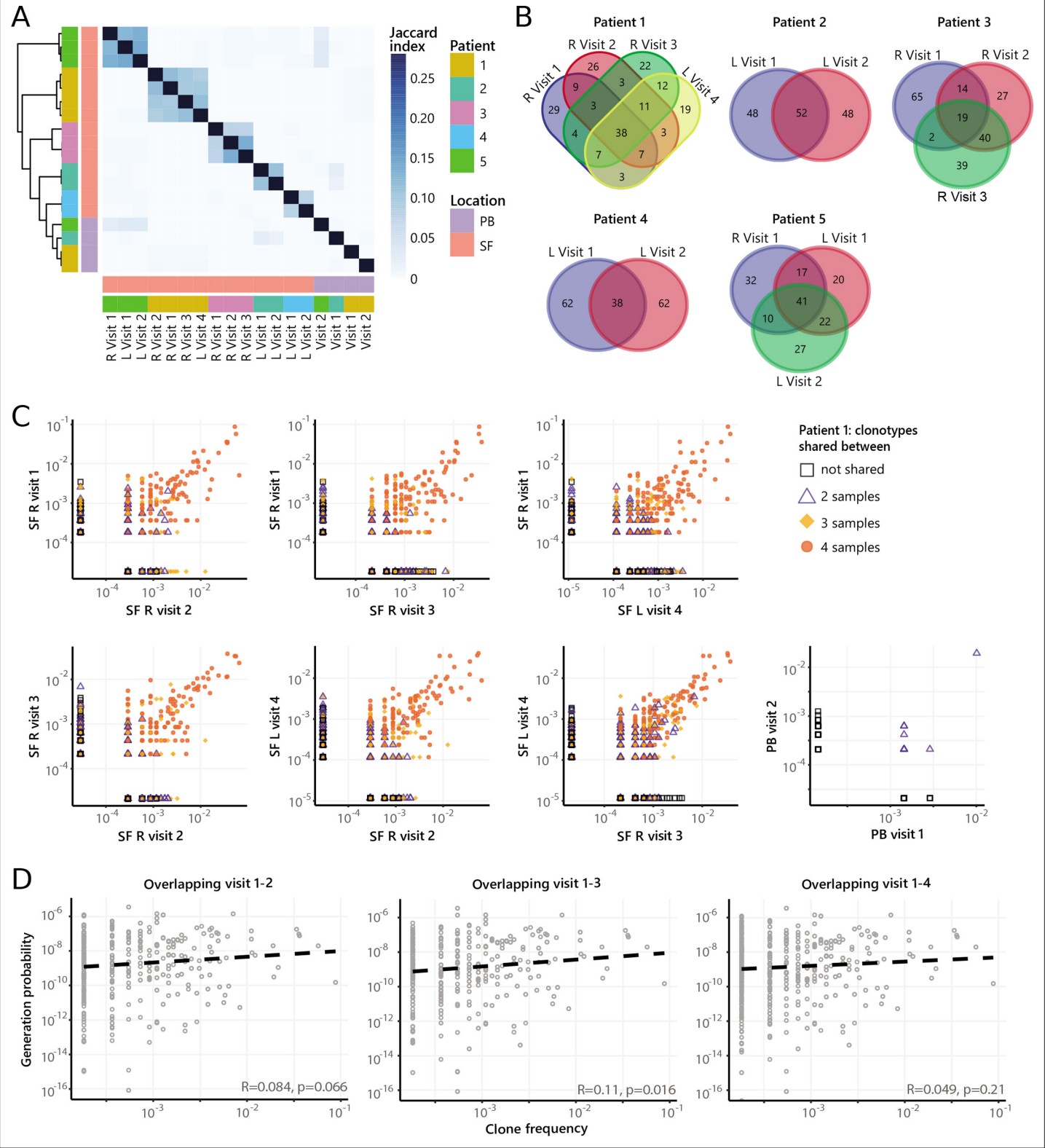

**Figure 5.** Persistence of non-Treg clones over the course of relapse remitting disease. (**A**) Heatmap showing overlap (Jaccard index, light blue = limited overlap, darkblue = high overlap) of non-Treg derived TCRβ sequences obtained from SF or PB from JIA patients over time. L, left knee, R, right knee. (**B**) Venn diagrams displaying the 100 most abundant unique TCRβ clones, defined by amino acid sequence, for longitudinal SF samples from all patients. (**C**) Frequency plots showing the overlapping non-Treg clones between visits for SF and PB, with color coding and shapes highlighting the number of samples in which unique clones are found. L, left; R, right. (**D**) Correlation (linear regression, dashed line) between frequency (x-axis) and

*Figure 5 continued on next page*

*Figure 5 continued*

generation probability (y-axis) of TCR clones shared across two visits for SF samples. JIA, juvenile idiopathic arthritis; PB, peripheral blood; SF, synovial fluid; TCR, T cell receptor.

The online version of this article includes the following figure supplement(s) for figure 5:

**Figure supplement 1.** JIA non-Treg TCRβ frequencies over time in the remaining four patients.

(1:2000 in 1.6% w/v paraformaldehyde, Fluidigm) overnight at 4°C or for 20 min on ice. Finally, the cells were washed and re-suspended with EQ Four Element Calibration beads (1:10, Fluidigm) at a concentration of $1\times10^6$ cells/ml. The cell mixture was then loaded and acquired on a Helios mass cytometer (Fluidigm) calibrated with CyTOF Tunning solution (Fluidigm). The output FCS files were randomized and normalized with the EQ Four Element Calibration beads (Fluidigm) against the entire run, according to the manufacturer's recommendations.

Normalized CyTOF output FCS files were de-barcoded manually into individual samples in FlowJo (v.10.2), and downsampled to equal cell events (5000 cells) for each sample. Batch run effects were assessed using an internal biological control (PBMC aliquots from the same healthy donor for every run). Normalized cells were then clustered with MarVis (*Kaever et al., 2009*), using Barnes Hut Stochastic Neighbor Embedding (SNE) nonlinear dimensionality reduction algorithm and k-means clustering algorithm, as previously described (*Chew et al., 2019*). The default clustering parameters were set at perplexity of 30, and p<1e−21. The cells were then mapped on a two-dimensional t-distributed SNE scale based on the similarity score of their respective combination of markers, and categorized into nodes (k-means). To ensure that the significant nodes obtained from clustering were relevant, we performed back-gating of the clustered CSV files and supervised gating of the original FCS files with FlowJo as validation. Visualizations (density maps, node frequency fingerprint, node phenotype, and radar plots) were performed through R scripts and/or Flow Jo (v.10.2). Correlation matrix and node heatmaps were generated using MarVis (*Kaever et al., 2009*) and PRISM (v.7.0).

## Single-cell RNA-sequencing analysis

Single-cell RNA-sequencing data from RA and OA SF was obtained from *Zhang et al., 2019*. We only included data from cells that were characterized as T cells. Next, we discarded cells with fewer than 1000 genes detected with at least one fragment, and cells that had more than 25% of reads from mitochondrial genes. Gene expression across cells was normalized using the *NormalizeData* function implemented in the R package Seurat (*Hao et al., 2021*) using the following parameters: *normalization.method = "LogNormalize", scale.factor=1e4*. Dimensionality reduction was performed using the Seurat *RunUMAP* function using the first 15 principal components. Clusters were annotated using the Seurat *FindClusters* function, with a resolution of 0.25. Differentially expressed genes between RA and OA patients were identified using the Seurat *FindMarkers* function with default parameters.

## TCR sequencing and analysis

Tregs and non-Tregs were lysed in RLT buffer (Qiagen, Hilden, Germany) and frozen at –80°C. Between $0.15\times10^6$ and $1\times10^6$ Tregs, and between $0.46\times10^6$ and $1\times10^6$ non-Tregs were obtained for TCR sequencing. Total RNA was isolated using the RNeasy Mini Kit (Qiagen) for cell fractions $\geq0.2\times10^6$ cells and the RNeasy Micro Kit (Qiagen) for fractions $\leq0.2\times10^6$ cells, following the manufacturer's instructions. cDNA was synthesized using the SMARTer RACE cDNA Amplification kit (Clontech, Palo Alto, CA). Amplification of the TCRβ VDJ region was performed using previously described primers and amplification protocols (*Zhou et al., 2006*). PCR product fragment size was analyzed using the QIAxcel Advanced System (Qiagen). End repair and barcode adapter ligation were performed with the NGSgo-LibrX and NGSgo-IndX (GenDx, Utrecht, The Netherlands) according to the manufacturer's instructions. Cleanup of the samples was performed after each step using High-Prep PCR beads and following the manufacturer's instructions (GC Biotech, Waddinxveen, The Netherlands). Paired-end next-generation sequencing was performed on the Illumina MiSeq system 500 (2×250 bp) (Illumina, San Diego, CA). TCR sequencing analysis was performed using RTCR as previously described (*Gerritsen et al., 2016*).

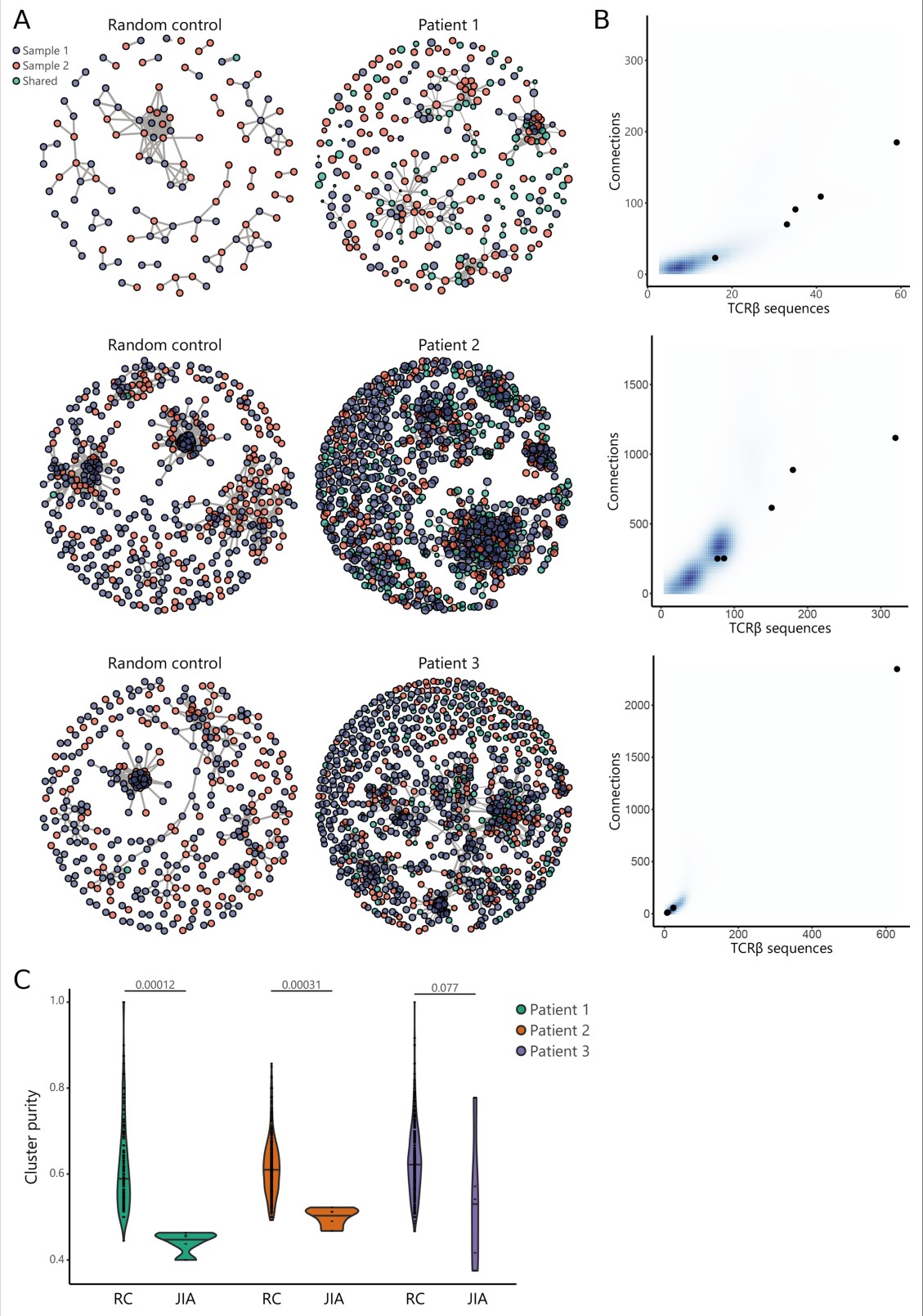

**Figure 6.** TCR similarity analysis of sequences found across distinct JIA patient knees. (**A**) TCR similarity networks based on amino acid k-mer sharing (k=3) between TCR sequences. Every node represents one TCRβ sequence, with sequences present in one sample (SF from left or right knees) highlighted in blue and orange, and sequences shared across two samples highlighted in green. Nodes are connected if TCRs share at least eight k-mers. Networks from JIA patient repertoires (right) are compared to random repertoires (left), with the same repertoire size. (**B**) Number of TCR

*Figure 6 continued on next page*

*Figure 6 continued*

sequences (x-axis) and their connections (y-axis) to other TCR sequences of the top five similarity clusters identified in (**A**). Blue density maps depict clusters identified in random repertoires (N=100), while black circles depict clusters identified in JIA patients. (**C**) Cluster purity (y-axis, %) for the top five clusters identified in random repertoires (RC), and JIA patient TCR similarity networks. Numbers indicate p-value of difference between RC and JIA (Mann-Whitney). JIA, juvenile idiopathic arthritis; RC, random repertoires; SF, synovial fluid; TCR, T cell receptor.

Healthy Treg TCRβ data (*Kasatskaya et al., 2020*) was obtained from the Gene Expression Omnibus (GEO). Diversity estimates were calculated by sample-size-based rarefaction and extrapolation using the R package iNEXT (iNterpolation/EXTrapolation) (*Hsieh et al., 2016*).

## TCR network analysis

For sequence similarity analysis, we counted the presence of overlapping 3-mer amino acid segments (defined as k-mers) in the TCRβ (CDR3) sequences. TCR sequences were considered similar when they shared at least eight k-mers, independent of the total sequence length. Random repertoires were generated using the generative model of V(D)J recombination implemented in OLGA (*Sethna et al., 2019*). For equal comparison to biological samples, random repertoires were downsampled to equal the number of TCR sequences. Cluster purity was calculated as the ratio of number of TCR sequences from the most abundant sequence within the cluster and the total number of TCR sequences in the cluster.

## Statistical analyses

Nonparametric Mann-Whitney (two-tailed) statistical test was performed in the manual gating of cellular subsets in FlowJo; p values<0.05 were considered statistically significant. The correlation matrix for the node frequency was calculated using Spearman's rank-order correlation. Generation probabilities ($p_{gens}$) of TCRβ amino acid sequences were computed using OLGA (*Sethna et al., 2019*). Figures were produced using the R package ggplot2 (*Wickham, 2016*). Venn diagrams were made on: http://bioinformatics.psb.ugent.be/webtools/Venn/.

## Acknowledgements

F van Wijk is supported by a VIDI Grant from ZonMw (91714332). AP is supported by Netherlands Organisation for Scientific Research (NWO) (Grant number 016.Veni.178.027).

## Additional information

### Funding

| Funder | Grant reference number | Author |
| --- | --- | --- |
| ZonMw | 91714332 | Femke van Wijk |
| Netherlands Organisation for Scientific Research | 016.Veni.178.027 | Aridaman Pandit |

The funders had no role in study design, data collection and interpretation, or the decision to submit the work for publication.

### Author contributions

Gerdien Mijnheer, Conceptualization, Formal analysis, Validation, Investigation, Visualization, Writing – original draft, Writing – review and editing; Nila Hendrika Servaas, Data curation, Formal analysis, Visualization, Writing – original draft, Writing – review and editing; Jing Yao Leong, Arjan Boltjes, Phyllis Chen, Formal analysis, Investigation, Writing – review and editing; Eric Spierings, Liyun Lai, Alessandra Petrelli, Sebastiaan Vastert, Investigation, Writing – review and editing; Rob J de Boer, Conceptualization, Investigation, Writing – review and editing; Salvatore Albani, Conceptualization, Resources, Investigation, Writing – review and editing; Aridaman Pandit, Conceptualization, Formal analysis, Supervision, Investigation, Visualization, Methodology, Writing – original draft, Writing – review and editing; Femke van Wijk, Conceptualization, Resources, Data curation, Supervision,

Funding acquisition, Investigation, Methodology, Writing – original draft, Project administration, Writing – review and editing

**Author ORCIDs**
Nila Hendrika Servaas ⬤ http://orcid.org/0000-0002-9825-7554
Eric Spierings ⬤ http://orcid.org/0000-0001-9441-1019
Rob J de Boer ⬤ http://orcid.org/0000-0002-2130-691X
Aridaman Pandit ⬤ http://orcid.org/0000-0003-2057-9737
Femke van Wijk ⬤ http://orcid.org/0000-0001-8343-1356

**Ethics**
Human subjects: Informed consent was obtained from all patients either directly or from parents/guardians when the patients were younger than 12 years of age. The study was conducted in accordance with the Institutional Review Board of the University Medical Center Utrecht (approval no. 11-499/C), in compliance with the Declaration of Helsinki.

**Decision letter and Author response**
Decision letter https://doi.org/10.7554/eLife.79016.sa1
Author response https://doi.org/10.7554/eLife.79016.sa2

## Additional files

**Supplementary files**
• Supplementary file 1. Clinical characteristics of JIA patients and healthy controls included in the study.
• Supplementary file 2. Overview of the CyTOF T cell panel with 37 markers.
• MDAR checklist

**Data availability**
TCR-sequencing data presented in this study have been deposited in NCBI's Gene Expression Omnibus (GEO) database under GSE196301. Both raw data and processed data are available.

The following dataset was generated:

| Author(s) | Year | Dataset title | Dataset URL | Database and Identifier |
| --- | --- | --- | --- | --- |
| Mijnheer G, Servaas NH, Yao Leong J, Boltjes A, Spierings E, Chen P, Lai L, Petrelli A, Vastert S, de Boer RJ, Albani S, Pandit A, van Wijk F | 2022 | Compartmentalization and persistence of dominant (regulatory) T cell clones indicates antigen skewing in juvenile idiopathic arthritis | http://www.ncbi.nlm.nih.gov/geo/query/acc.cgi?acc=GSE196301 | NCBI Gene Expression Omnibus, GSE196301 |

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
