## [Editor Report]

In this study, the authors performed mass cytometry analysis and T cell receptor sequencing to study different joints affected at the same time and over the course of the relapsing-remitting disease and found that the composition and functional characteristics of immune infiltrates are similar between joints within one Juvenile Idiopathic Arthritis patient. They observed an overlap between dominant T cell clones, especially Treg, and some of these clones could be detected in circulation and persisted over the course of the relapsing-remitting disease. They demonstrated that these T cell clones were characterized by a high degree of sequence similarity, indicating the presence of TCR clusters responding to the same antigens.

---

## [Decision Letter]

**Decision letter after peer review:**

Thank you for submitting your article "Compartmentalization and persistence of dominant (regulatory) T cell clones indicates antigen skewing in juvenile idiopathic arthritis" for consideration by *eLife*. Your article has been reviewed by 2 peer reviewers, and the evaluation has been overseen by a Reviewing Editor and Mone Zaidi as the Senior Editor. The reviewers have opted to remain anonymous.

Essential revisions:

1. The criteria for the JIA patient's recruitment should be clearly presented in the method section.

2. Is it is possible to obtain healthy SFMC for CyTOF analysis or TCR sequencing?

3. In figure 3, the authors need to test T cell clones in the healthy PBMC and compare T cell clones from healthy and JIA PBMCs.

4. In Supplementary Figure 3, the interval of PB and SF sample collections is not consistent. Only 1 patient completed 4 visits with the sample collections. It is difficult to make conclusion based on the data obtained from 1 patient. The authors need to increase patient numbers.

*Reviewer #1 (Recommendations for the authors):*

1. The mass spectrometry flow data in this paper showed that there was heterogeneity among the patients, but only three patients were tested in this paper. Is it necessary to increase the number of patients?

2. If it is possible to obtain healthy SFMC for CyTOF analysis or TCR sequencing?

3. In figure 3, the authors need to test T cell clones in the healthy PBMC. Please compare T cell clones from healthy PBMC with those from JIA PBMC.

*Reviewer #2 (Recommendations for the authors):*

The author should justify the different patient numbers in several experiments and include more patients for certain experiments. It will also help if the authors can collect the samples from different joints to solidify their findings since JIA occurs in several joints.

---

## [Author Response]

Essential revisions:1. The criteria for the JIA patient's recruitment should be clearly presented in the method section.

A total of 9 JIA patients were included in this study. Of these, n=2 were diagnosed with extended oligo JIA, n=2 with rheumatoid factor negative poly-articular JIA, and n=5 with oligo JIA, according to the revised criteria for JIA. The average age at the time of inclusion was 13,1 years (range 3,2 – 18,1 years) with a disease duration of 7,3 years (range 0.4 – 14.2 years). Due to limited sample availability, we did not have strict inclusion or exclusion criteria for JIA patient recruitment. For CyTOF analysis, patients were selected based on the criteria that the left and right knee joints should both be affected at the time of inclusion. For sequential TCR sequencing analysis, we included patients with a refractory disease course. At the time of first inclusion, patients were treated with non-steroidal anti-inflammatory drugs (NSAIDs) or methotrexate, but no biologicals. For the refractory time point samples, patients were treated with disease modifying anti-rheumatic drug (DMARDs) (leflunomide) and/or biologicals (humira) after first sample inclusion due to the refractory nature of their disease. Despite some heterogeneity within the patient population, similar patterns were observed in all patients, both for the spatial and over time analysis. This suggests that the presented concepts of spatial overlap of dominant T cell clones, and (re)expansion of dominant clones over time with disease flares are robust and not confounded by e.g. treatment or JIA subclass.

We have now updated the methods section (lines 455-463) of the revised manuscript with more information on patient recruitment, and included information on diagnosis, sex, age, disease duration and medication for every patient in Supplementary File 1.

2. Is it is possible to obtain healthy SFMC for CyTOF analysis or TCR sequencing?

Unfortunately, obtaining SFMCs from healthy joint tissue is not possible as joint aspirates are not taken from healthy donors and, most importantly, the fluid does not contain enough cells to perform TCR-seq on the scale that is required to reliably answer our questions. To our knowledge, there is also no public TCR or CyTOF data from healthy joint biopsies available in the current literature. As a proxy for healthy samples, autoimmune rheumatic samples are often compared to samples from patients suffering from osteoarthritis (OA), a degenerative, non-autoimmune disease of the joints. Therefore, we sought to validate our findings in publicly available datasets comparing SFMCs from autoimmune rheumatic samples to non-autoimmune OA samples. To this end, we reanalyzed single-cell RNA-sequencing data obtained from Zhang, et al. (Zhang, F., et al. Nat. Immunol. 20, 928–942 (2019). https://doi.org/10.1038/s41590-019-0378-1) and compared the gene expression profiles of SF specific T cells between RA and OA patients.

In this data, we identified a cluster of CD4^+^FOXP3+ Tregs (characterized by high CD4, FOXP3, PDCD1, CTLA4, TIGIT and IL2RA expression, Figure 2—figure supplement 2A and B) that showed increased frequency in RA patients (Figure C attached on the next page), consistent with the high frequency of Tregs that we observed in our JIA SFMC samples. Additionally, the expression of markers of chronic TCR activation (PDCD1 (PD1), CTLA4 and ICOS), and cytokines (TNF, IFNΓ and GZMB) were significantly increased in RA compared to OA, in line with what we observed in JIA SFMC (Figure D attached on the next page). These findings also match previous transcriptome analyses, which showed that Tregs from synovial fluid of RA patients display a very similar gene expression signature as Tregs from JIA patients (Mijnheer, G., et al. Nat. Commun. 12, 2710 (2021). https://doi.org/10.1038/s41467-021-22975-7). These results further highlight that Tregs in rheumatic autoimmune disease are characterized by inflammation related molecular signatures potentially relevant for autoimmune disease. Further validation of our observations in larger cohorts of JIA patients should help to substantiate these results and aid the identification of pathogenic Treg populations across patients.

We have now added this new analysis to our manuscript in Figure 2—figure supplement 2, lines 183-198 (Results section), and lines 362-371 (Discussion section).

3. In figure 3, the authors need to test T cell clones in the healthy PBMC and compare T cell clones from healthy and JIA PBMCs.

We obtained publicly available TCRβ sequencing data of Tregs (CD3^+^CD4^+^CD25^high^ CD127^low^) sorted from healthy donor PBMCs (GSE158848, S.A. Kasatskaya. et al. *eLife* 9:e57063 (2020). https://doi.org/10.7554/eLife.57063). To compare the diversity of the Treg repertoires between healthy controls (HC) and JIA patients, we performed rarefaction analysis. The estimated species richness and Shannon diversity was significantly lower for JIA patients than HC (see figure 3—figure supplement 1), demonstrating that the JIA Treg repertoire is less diverse than healthy.

These observations also match previous findings that the diversity of the Treg TCR repertoire in JIA patients is very low, and increases after HSCT (E.M. Delemarre, et al. Blood 2016. https://doi.org/10.1182/blood-2015-06-649145).

These new analyses have now been included in our revised manuscript in Figure 3—figure supplement 1, lines 249-255 (Results sections) and lines 390-393 (Discussion section).

4. In Supplementary Figure 3, the interval of PB and SF sample collections is not consistent. Only 1 patient completed 4 visits with the sample collections. It is difficult to make conclusion based on the data obtained from 1 patient. The authors need to increase patient numbers.

In our study, we had longitudinal samples available for 5 JIA patients for which we performed TCR sequencing of Tregs from SFMCs from different joints (right or left) at at least two time points. In the manuscript we mainly focused on patient 1, as for this patient the largest amount of data/timepoints were available. However, for all other longitudinal patient samples included, we also show that dominant clones persist over time (figure 4A and 5A). To further highlight that our observations are not just applicable to one patient, but consistent for all patients included, we now included detailed analysis for all patients in new Figure 4—figure supplement 3 and Figure 5—figure supplement 1. This analysis shows that frequencies of shared TCRβs are consistent over time in all patients.

Reviewer #1 (Recommendations for the authors):1. The mass spectrometry flow data in this paper showed that there was heterogeneity among the patients, but only three patients were tested in this paper. Is it necessary to increase the number of patients?

Indeed, there is a degree of heterogeneity observable between the different patients included in our study. However, within each patient, the mass spectrometry data revealed nearly identical profiles for the left and right knee joints. Thus, although inter-individual heterogeneity can be observed, intra-individual differences are very minimal. This is why in the manuscript, we mainly focus on observations that are consistent within individual patients rather than across patients.

However, we agree with the reviewer that inter-patient heterogeneity is present and warrants validations of our results in bigger patient cohorts. Therefore, we have now added the following line in the discussion to highlight this (lines 369-371): “Further validation of our observations in larger cohorts of JIA patients should help to substantiate these results and aid the identification of pathogenic Treg populations across patients.”.

In addition, in order to strengthen our observations, we now also included single-cell transcriptomics data obtained from Zhang, et al. (Zhang, F., et al. Nat. Immunol. 20, 928–942 (2019). https://doi.org/10.1038/s41590-019-0378-1). In this new analysis, we identified a cluster of CD4^+^FOXP3+ Tregs (new Figure 2—figure supplement 2A/B) that showed increased frequency in RA patients (new Figure 2—figure supplement 2C), consistent with the high frequency of Tregs that we observed in our JIA SFMC samples. Additionally, the expression of markers of chronic TCR activation (PDCD1 (PD1), CTLA4 and ICOS), and cytokines (TNF, IFNΓ and GZMB) were significantly increased in RA compared to OA, in line with what we observed in JIA SFMC (new Figure 2—figure supplement 2D). These results demonstrate that, although the number of JIA patients included in our study is low, obtained results are robustly reproducible in an independent, comparable dataset.

2. If it is possible to obtain healthy SFMC for CyTOF analysis or TCR sequencing?

Unfortunately, obtaining SFMCs from healthy joint tissue is not possible as joint aspirates are not taken from healthy donors and, most importantly, the fluid does not contain enough cells to perform TCR-seq and CyTOF on the scale that is required to reliably answer our questions. To our knowledge, there is also no public TCR or CyTOF data from healthy joint biopsies available in the current literature. As a proxy for healthy samples, rheumatic samples are often compared to samples from patients suffering from osteoarthritis (OA), a degenerative, non-autoimmune disease of the joints. Therefore, we sought to validate our findings in publicly available datasets comparing SFMCs from autoimmune rheumatic samples to non-autoimmune OA samples. To this end, we reanalyzed single-cell RNA-sequencing data obtained from Zhang, et al. (Zhang, F., et al. Nat. Immunol. 20, 928–942 (2019). https://doi.org/10.1038/s41590-019-0378-1) and compared the gene expression profiles of SF specific T cells between RA and OA patients.

In this data, we identified a cluster of CD4^+^FOXP3+ Tregs (characterized by high CD4, FOXP3, PDCD1, CTLA4, TIGIT and IL2RA expression, new Figure 2—figure supplement 2A/B) that showed increased frequency in RA patients (new Figure 2—figure supplement 2C), consistent with the high frequency of Tregs that we observed in our JIA SFMC samples. Additionally, the expression of markers of chronic TCR activation (PDCD1 (PD1), CTLA4 and ICOS), and cytokines (TNF, IFNΓ and GZMB) were significantly increased in RA compared to OA, in line with what we observed in JIA SFMC (new Figure 2—figure supplement 2D). These findings also match previous transcriptome analyses, which showed that Tregs from synovial fluid of RA patients display a very similar gene expression signature as Tregs from JIA patients (Mijnheer, G., et al. Nat. Commun. 12, 2710 (2021). https://doi.org/10.1038/s41467-021-22975-7). These results further highlight that Tregs in rheumatic autoimmune disease are characterized by inflammation related molecular signatures potentially relevant for autoimmune disease. Further validation of our observations in larger cohorts of JIA patients should help to substantiate these results and aid the identification of pathogenic Treg populations across patients.

We have now added this new analysis to our manuscript in new Figure 2—figure supplement 2, lines 183-198 (Results section), and lines 362-371 (Discussion section).

3. In figure 3, the authors need to test T cell clones in the healthy PBMC. Please compare T cell clones from healthy PBMC with those from JIA PBMC.

We obtained publicly available TCRβ sequencing data of Tregs (CD3^+^CD4^+^CD25^high^ CD127^low^) sorted from healthy donor PBMCs (GSE158848, S.A. Kasatskaya. et al. *eLife* 9:e57063 (2020). https://doi.org/10.7554/eLife.57063). To compare the diversity of the Treg repertoires between healthy controls (HC) and JIA patients, we performed rarefaction analysis. The estimated species richness and Shannon diversity was significantly lower for JIA patients than HC (Figure 3—figure supplement 1), demonstrating that the JIA Treg repertoire is less diverse than healthy.

These observations also match previous findings that the diversity of the Treg TCR repertoire in JIA patients is very low, and increases after HSCT (E.M. Delemarre, et al. Blood 2016. https://doi.org/10.1182/blood-2015-06-649145).

These new analyses have now been included in our revised manuscript in new Figure 3—figure supplement 1, lines 249-255 (Results sections) and lines 390-393 (Discussion section).